# Evaluating the Impact of Multimodal Prehabilitation with High Protein Oral Nutritional Supplementation (HP ONS) with Beta-Hydroxy Beta-Methylbutyrate (HMB) on Sarcopenic Surgical Patients—Interim Analysis of the HEROS Study

**DOI:** 10.3390/nu16244351

**Published:** 2024-12-17

**Authors:** Frederick Hong-Xiang Koh, Vanessa Yik, Shuen-Ern Chin, Shawn Shi-Xian Kok, Hui-Bing Lee, Cherie Tong, Phoebe Tay, Esther Chean, Yi-En Lam, Shi-Min Mah, Li-Xin Foo, Clement C Yan, Wei-Tian Chua, Haziq bin Jamil, Khasthuri G, Lester Wei-Lin Ong, Alvin Yong-Hui Tan, Koy-Min Chue, Leonard Ming-Li Ho, Cheryl Xi-Zi Chong, Jasmine Ladlad, Cheryl Hui-Min Tan, Nathanelle Ann Xiaolian Khoo, Jia-Lin Ng, Winson Jianhong Tan, Fung-Joon Foo

**Affiliations:** 1Colorectal Service, Sengkang General Hospital, Singapore 544886, Singapore; 2Duke-NUS Medical School, Singapore 169857, Singapore; 3Lee Kong Chian School of Medicine, Singapore 308232, Singapore; 4Department of Radiology, Sengkang General Hospital, Singapore 544886, Singapore; shawn.kok.s.x@singhealth.com.sg (S.S.-X.K.);; 5Department of Dietetics, Sengkang General Hospital, Singapore 544886, Singapore; 6Department of Physiotherapy, Sengkang General Hospital, Singapore 544886, Singapore; 7Department of General Surgery, Sengkang General Hospital, Singapore 544886, Singapore

**Keywords:** prehabilitation, oral nutritional supplementation, beta-hydroxy beta-methylbutyrate, sarcopenia, intramuscular adipose tissue

## Abstract

Background: Multimodal prehabilitation programs, which may incorporate nutritional supplementation and exercise, have been developed to combat sarcopenia in surgical patients to enhance post-operative outcomes. However, the optimal regime remains unknown. The use of beta-hydroxy beta-methylbutyrate (HMB) has beneficial effects on muscle mass and strength. However, its effect on muscle quality in the perioperative setting has yet to be established. This study aims to explore the impact of a multimodal prehabilitation program using a bundle of care that includes high-protein oral nutritional supplementation (HP ONS) with HMB and resistance exercise on muscle quality and functional outcomes in sarcopenic surgical patients. Methods: Sarcopenic adult patients undergoing elective major gastrointestinal surgeries were recruited for this pilot interventional cohort study. They were enrolled in a 2–4-week multimodal prehabilitation program comprising resistance exercise, nutritional supplementation, vitamin supplementation, comorbid optimization and smoking cessation. Participants were provided three units of HP ONS with HMB per day pre-operatively. The primary outcome was changes in intramuscular adipose tissue (IMAT) as a proxy of muscle quality, assessed using Artificial Intelligence (AI)-aided ultrasonography. Secondary outcomes include changes in anthropometric measurements and functional characteristics. Outcomes were measured before prehabilitation, after prehabilitation and 1 month post-operatively. Results: A total of 36 sarcopenic patients, with a median age of 71.5 years, were included in this study. There was an increase in the IMAT index after two weeks of prehabilitation (*p* = 0.032) to 1 month after surgery (*p* = 0.028). Among functional parameters, improvement was observed in gait speed (*p* = 0.01) after two weeks of prehabilitation, which returned to baseline post-operatively. The median length of hospital stay was 7 (range: 2–75) days. Conclusions: The increase in the IMAT index in a sarcopenic surgical cohort undergoing prehabilitation may be due to altered muscle metabolism in elderly sarcopenic patients. A prehabilitation regime in sarcopenic patients incorporating HP ONS with HMB and resistance exercise is feasible and is associated with increased gait speed.

## 1. Introduction

Sarcopenia is a condition characterized by the loss of skeletal muscle mass and function [1], with an estimated prevalence of more than 30% among older adults in Singapore [2]. This prevalence increases in acute hospital patients and has been associated with increased morbidity and mortality [3,4,5]. Sarcopenia puts patients at greater risk of adverse outcomes after surgery, and multiple studies have also linked sarcopenia to increased incidences of post-operative complications [6]. Hence, prehabilitation programs before major surgery have been trialed to improve post-surgical outcomes with promising results [7,8,9,10].

Prehabilitation refers to a wide range of interventions that may be implemented before a patient undergoes surgery, with the aim to build up a patient’s physiological reserves to aid post-operative recovery and reduce morbidity [7]. Interventions include exercise, nutrition support, comorbidity optimization, smoking and alcohol cessation, respiratory interventions, psychological support, and patient education [7,8,9,10,11]. Multimodal prehabilitation programs incorporate several of the above elements to achieve a synergistic effect in optimizing pre-operative fitness, with nutritional supplementation and exercise at the core of most programs.

Nutritional supplementation with protein and leucine has beneficial effects on lean body mass and skeletal muscle mass, improving muscle function [12,13,14]. In particular, beta-hydroxy beta-methylbutyrate (HMB), a metabolite of the essential branched-chain amino acid (BCAA) leucine, has been reported to have anti-inflammatory effects on muscle, inhibit protein degradation and promote anabolic effects on protein metabolism [15]. Studies have suggested its use in increasing mass and enhancing strength of skeletal muscles in the elderly [16,17,18]. Studies in orthopedic [19] and geriatric [20] patients have reported improved post-operative outcomes and muscle function with the use of HMB supplementation. However, these benefits have not been well studied among patients with sarcopenia undergoing gastrointestinal surgery. Optimizing nutrition with a high-protein, high-calorie oral nutritional supplementation (HP ONS) with HMB provides increased amounts of macro- and micronutrients to meet the increased needs in patients undergoing prehabilitation with exercise. This could help combat the increased risk of poor post-operative outcomes for sarcopenic patients undergoing major gastrointestinal surgery [21].

Despite increasing awareness and a growing body of research, the optimal prehabilitation regime and duration is still unknown. Furthermore, the window for intervention may be limited—particularly in oncologic surgery—as the risks and benefits of delaying surgery for prehabilitation must be considered. A duration of at least eight weeks is typically required for appreciable increase in muscle mass [22]. Yet, published prehabilitation protocols often range from two to four weeks long [8,10,11]. Due to this relatively short prehabilitation process, changes in muscle mass and functional outcomes in sarcopenic patients may be limited and difficult to assess.

Sarcopenia affects muscle composition, resulting in reduced muscle mass and increased adipose tissue [23]. A key component of sarcopenia diagnosis is a decrease in muscle quantity, assessed by measuring appendicular skeletal muscle mass in diagnostic guidelines [24]. However, muscle quality is also emerging as an important aspect of sarcopenia. Changes in muscle metabolism occur with age due to decline in skeletal muscle mitochondrial function and may underlie the decline in muscle health and performance during aging [25]. Myosteatosis, defined as skeletal muscle fat infiltration, is a marker of muscle quality and correlates with muscle strength and health [26,27]. Myosteatosis can be reflected by measuring intramuscular adipose tissue (IMAT)—the accumulation of fat within muscle fibers—via various imaging modalities [28,29,30]. It is possible that alterations in muscle quality, i.e., IMAT, may precede appreciable changes in muscle mass but correlate with functional outcomes in the short prehabilitation window [31,32].

Hence, this pilot study aims to evaluate the effect of a multimodal prehabilitation program using an HP ONS with HMB and resistance exercise on muscle quality and functional outcomes in sarcopenic patients undergoing gastrointestinal surgery. By elucidating the importance of HP ONS with HMB and resistance exercise, this study adds to the body of literature to help uncover an optimal prehabilitation regime that could benefit more patients in the future. This study is novel in its use of IMAT and its derivatives to evaluate muscle quality in a multimodal prehabilitation program including HP ONS with HMB supplementation and resistance exercise.

## 2. Methods

### 2.1. Study Design

A prospective non-randomized interventional pilot cohort study was performed in a single institution to evaluate the effect of a multimodal prehabilitation program using an HP ONS with HMB in sarcopenic patients undergoing surgery. This is an interim analysis of the study protocol titled “Oral Nutritional Supplementation with **H**MB **e**nhance muscle quality in sa**r**c**o**penic **s**urgical patients (HEROS)—a pilot interventional cohort study”. This study was approved by the SingHealth Centralized Institute Review Board (CIRB: 2022/2027) and registered with https://clinicaltrials.gov/ (NCT05344313). A total of 40 patients were recruited for this proof-of-concept study. The number of participants derived was arbitrary as part of a proof-of-concept. Patients included in this study were adult patients between 40 and 90 years old, sarcopenic, due to undergo elective major gastrointestinal surgery, ambulant and were able to comply with physiotherapy and dietitian advice. Patients who were pregnant, unable to provide informed consent, unable to answer questionnaires, declined or could not be assessed for sarcopenia were excluded from this study. Other exclusion criteria were patients with disease conditions requiring emergent or semi-emergent operation, diabetes mellitus, and chronic kidney disease or end stage renal failure due to their medical unsuitability to consume the studied HP ONS.

Eligible patients were recruited from Sengkang General Hospital from June 2022 to January 2024, and written informed consent was obtained. The patients were evaluated for sarcopenia based on the Asian Working Group for Sarcopenia 2019 guidelines (AWGS 2019) [24]. The multimodal prehabilitation regime in our study consisted of oral nutritional supplementation, resistance training exercises, vitamin D and iron supplementation, comorbidity optimization and smoking cessation.

Ensure^®^ Plus Advance with HMB (Abbott Nutrition, Chicago, IL, USA) was used as the high-protein oral nutritional supplement in our study, aiming for a target 120% of daily caloric and protein requirement to account for increased activity and stress factors. Patients were prescribed 3 units of Ensure^®^ Plus Advance to consume daily prior to surgery for a prehabilitation duration of between two to four weeks, and 2 units of Ensure^®^ Plus Advance to consume daily after surgery for two months (Figure 1). The macronutrient breakdown is shown in Table 1. Compliance was assessed through a self-administered compliance diary and review of leftover stock by a study-dedicated dietitian. The exercise protocol involved resistance training administered by a study-dedicated physiotherapist on an outpatient basis. Following an initial physiotherapy assessment of strength, balance, walking speed and cardiovascular endurance, patients were prescribed individualized resistance exercises based on their ability. Exercises included both upper and lower limb exercises in seated and standing positions. Resistance exercises were usually performed using a resistance band, dumbbells, weighted objects such as water bottles, or with their own body weight if they were unable to perform the exercises safely and in good form. Examples of resistance exercises included seated knee extension, standing hip abduction, seated horizontal pulls and standing bent rows. One to three sets of 10–15 repetitions were prescribed for each exercise, for a total of 8–10 exercises that targeted major muscle groups to be completed five times a week. A review of exercise technique was performed by the physiotherapist at a one-week review, where the exercises could be progressed accordingly. Vitamin D supplementation was given as 50,000 IU once a week for eight weeks, and iron supplementation was given as 1 g of ferric derisomaltose intravenously once from Week 0.

A total of 10 healthy participants were also recruited to serve as a baseline comparison of muscle quality between sarcopenic and non-sarcopenic patients. Likewise, the sarcopenia status was assessed according to AWGS 2019, and muscle quality was assessed. These participants did not undergo prehabilitation or surgery.

### 2.2. Variables and Outcomes

Patient characteristics, such as the patient’s demographics, anthropometric measurements, and sarcopenia status according to AWGS 2019, were collected.

The assessment of intramuscular adipose tissue and its derivatives, as a proxy of muscle quality, was performed with ultrasonography of participants’ bilateral rectus femoris muscles using an Artificial Intelligence (AI)-aided program, MuscleSound^®^ (Denver, CO, USA) by a trained sonographer. Ultrasound images were obtained using the Philips Lumify ultrasound system (Philips, Amsterdam, The Netherlands) with a linear array transducer (4–12 MHz, 34 mm aperture size, Lumify L12-4 Android). The midpoint of the rectus femoris was scanned while the patient was lying supine on an examination bed, using standardized landmarks provided by MuscleSound^®^ based on the individual’s height. IMAT% was automatically calculated by MuscleSound^®^ from ultrasound echo intensity using the equation published by Young et al. [33], and the results were averaged between the left and right rectus femoris readings. The IMAT index (%/cm^2^), a derivative of IMAT%, was calculated by dividing IMAT% by muscle area (cm^2^) to correct for muscle size.

Secondary outcomes in this study were changes in functional parameters. These parameters include exercise tests by physiotherapists (handgrip strength, 30-sec chair rise, functional reach test, 6-min walk test (6MWT) and gait speed) and anthropometric measurements by dietitians (mid-arm circumference, mid-arm muscle circumference, mid-arm muscle area and triceps skinfold). The methods by which each exercise test was performed is described in Table 2. For each test, a trial of three tries was given. The surgical complications, length of stay in hospital and discharge disposition following surgery were also recorded to evaluate short-term postoperative outcomes.

Study outcomes were measured at baseline, prior to prehabilitation (“Week 0”), after two weeks of prehabilitation (“Week 2”) and one month after surgery (“Post-Op 1 Month”).

### 2.3. Statistical Analysis

All statistical analyses were performed using IBM SPSS Statistics for Macintosh Version 29.0.1 (IBM Corp, Armonk, NY, USA) and GraphPad Prism Version 10.1.1 (GraphPad Software, Boston, MA, USA). Continuous variables were presented as median with range. Categorical variables were presented as numbers and percentages (%). Box and whisker plots were plotted to visualize the changes in IMAT and functional outcomes over the peri-operative period, as well as to identify potential outliers. Outliers were defined as >1.5 standard deviations from the mean, and far outliers as >3 standard deviations from the mean. Outliers and far outliers were excluded from subsequent statistical analyses. Univariate analyses were performed using the Wilcoxon Signed Rank test, with *p* < 0.05 considered as statistically significant.

## 3. Results

### 3.1. Patient Characteristics

A total of 40 patients were recruited for the interventional cohort study. After applying the inclusion and exclusion criteria, 36 patients were eligible for further analysis. The 36 patients had a median age of 71.5 years, of which half were male (50%). The median BMI was 21.6 kg/m^2^ (12.9–28.4). A total of 24 (66.7%) patients had severe sarcopenia, while 12 (33.3%) had sarcopenia. The median compliance to HP ONS with HMB prescription was 83% (23.3–100%) among the eligible participants. The median length of stay was 7 (2–75) days. A total of 28 (77.8%) patients were discharged home, and 6 (16.7%) patients were transferred to a step-down rehabilitation facility. More details can be found in Table 3.

### 3.2. Outcomes

The study outcomes for muscle quality, functional parameters and anthropometric measurements are shown in Table 4. The median and standard deviation are reported at Week 0, Week 2 and Post-Op 1 Month.

There was a significant increase in the IMAT index (IMAT% divided by muscle area to correct for muscle size) during the prehabilitation period from Week 0 to Week 2 (median: 4.78 vs. 6.06, *p* = 0.032), sustained into the post-operative period from Week 0 to Post-Op 1 Month (median: 4.78 vs. 6.08, *p* = 0.028) (Figure 2). At Post-Op 1 Month, an increase in IMAT index was observed in 63.6% of all patients compared to Week 0. The patients were subdivided into patients below 65 years old and aged 65 years and above for further analysis. Patients aged 65 years and above had a more significant increase in the IMAT index through the perioperative period from Week 0 to Week 2 (median: 4.76 vs. 6.22, *p* = 0.011) and from Week 0 to Post-Op 1 Month (median: 4.76 vs. 6.09, *p* = 0.012) (Figure 3). At Post-Op 1 Month, an increase in the IMAT index was observed in 67.7% of patients aged 65 years and above. There was no significant change in IMAT% during the peri-operative period, from Week 0 to Week 2 (*p* = 0.837) and from Week 0 to Post-Op 1 Month (*p* = 0.908).

In terms of functional parameters, there was improvement in the 6MWT (median: 345 vs. 387, *p* = 0.005) and gait speed (median: 0.90 vs. 1.05, *p* = 0.01) from Week 0 to Week 2, but no significant changes in grip strength, 30 s chair rise and functional reach test. The 6MWT and gait speed outcomes returned to baseline after surgery, with no significant change from Week 0 to Post-Op 1 Month for all functional parameters (Figure 4).

There were also no significant changes in mid-arm circumference, mid-arm muscle circumference, mid arm muscle area and triceps skinfold through the peri-operative period (Figure 5). Weight and BMI were maintained in the prehabilitation period with no significant change from Week 0 to Week 2. There was a statistically significant decrease in weight (median: 50.8 vs. 51.2, mean: 51.1 vs. 50.6, *p* = 0.021) and BMI (median: 21.8 vs. 21.1, mean: 20.9 vs. 20.6, *p* = 0.023) from Week 0 to Post-Op 1 Month (Figure 6).

### 3.3. Baseline Non-Sarcopenic Muscle Characteristics

A total of ten healthy participants were recruited for muscle quality assessment (Table 5). The median age was 62.5 (range: 59–74), the median BMI was 24.4 (range: 19.0–33.2), and four subjects were male (40%). All the participants were not sarcopenic and did not have any chronic medical conditions. The median IMAT% was 14.04%, and the IMAT index was 4.25%/cm^2^.

## 4. Discussion

At the time of writing, this is the first study to evaluate the impact of multimodal prehabilitation incorporating HP ONS with HMB and resistance exercise on sarcopenic surgical patients and its effect on muscle quality. In this study, IMAT and its derivatives (IMAT index) were used as markers of muscle quality, considering its correlations with muscle health and sarcopenia [30,34]. The implementation of HP ONS with HMB and resistance exercise in prehabilitation was evaluated for its purported benefits in muscle building and improving muscle function to combat sarcopenic changes in the peri-operative period.

### 4.1. Prehabilitation Outcomes

The main finding was an increase in the IMAT index after a regime of HP ONS with HMB and resistance exercise for two weeks prior to surgery as part of a multimodal prehabilitation regime, which was sustained to one month post-operatively. The subgroup analysis of the IMAT index of patients aged 65 years and above yielded a lower *p*-value, suggesting a more significant increase in the IMAT index in older patients. Intramuscular adipose tissue is understood as a marker of, but unlikely to be a major cause of, muscle dysfunction [35]. Skeletal muscle shows a strong metabolic aging phenotype [36], and the alterations in muscle metabolism that occur with increasing age may explain the observed changes in the IMAT index after prehabilitation [25]. Compared to the healthy non-sarcopenic participants, the baseline IMAT index of sarcopenic patients is expectedly higher. In our study, there was no significant change in ultrasound-derived IMAT% throughout the peri-operative period. This suggests that IMAT% itself was not clinically useful in evaluating changes in muscle quality, and correction, such as IMAT index for muscle size, is needed [30,37].

Following two weeks of prehabilitation with HP ONS with HMB and resistance exercise, there was significant improvement in the 6MWT and gait speed. The 6MWT was used to derive gait speed, by dividing the distance in meters covered over 6 min. Gait speed is a measure of overall functional mobility and capacity [38,39], unlike other parameters that focus on specific muscle groups, such as grip strength, which measures the strength of the hand-finger segment [40]. A gait speed of more than 0.8 m/s is associated with community ambulation in adults [41], and a threshold of gait speed less than 1.0 m/s is generally accepted as a marker for increased fall risk [42]. Hence, the improvement in gait speed in this study from a median of 0.90 to 1.05 m/s in two weeks is clinically significant. However, there was return to baseline function in gait speed at one month after surgery. This may support the continued administration of HP ONS with HMB and resistance exercise in the post-operative rehabilitation period to sustain functional improvement. The other functional parameters and anthropometric measurements had no significant change over the peri-operative period. It was reassuring that there was no deterioration in functional parameters post-operatively compared to the pre-operative baseline, as some level of functional deterioration is typically expected in elderly patients undergoing major surgery [43]. This supports the utility of prehabilitation and rehabilitation in maintaining baseline function among patients.

Weight gain is a reported benefit of ONS intervention for adults at risk of malnutrition [44,45]. Changes in weight from positive energy balance over the short peri-operative period ranging from two to four weeks would predominantly be due to fat rather than lean tissue, which requires a longer time to see an appreciative change. Body fat serves as a reservoir of energy in catabolic states, and maintaining adequate body fat may be beneficial in patients before undergoing surgery [46]. While there was a statistically significant change in weight and BMI from the start of prehabilitation to post-surgery, these changes in the magnitude of several grams, may not be clinically significant. Additionally, the nature of surgery in some cases (e.g., the resection of diseased organ) may cause a decrease in weight after surgery.

In this study, the median length of stay was 7 days, which was lower than the average length of stay reported for most patients undergoing similar surgeries (ranging 8–11 days) [8,47], with a high rate of discharge to home, as opposed to additional rehabilitation at a step-down care facility. There was acceptable adherence to HP ONS with HMB during the peri-operative period, with compliance at 83% as assessed by the study dietitian.

### 4.2. Effect of HP ONS with HMB and Resistance Exercise on Muscle Quality

The positive impact of HMB supplementation on muscle strength and physical performance has been demonstrated in healthy and pre-frail older adults [48,49]. However, the duration of nutritional intervention in those studies ranged from 8 to 12 weeks and were conducted in participants with no underlying medical conditions. Therefore, we do not expect that two to four weeks of prehabilitation can reverse sarcopenic changes nor greatly increase muscle mass in our study’s patients. Rather, the aim of prehabilitation is to build up a patient’s physiological reserves to aid post-operative functional recovery [7]. During this prehabilitation period, there was interest in the changes in muscle metabolism and composition that take place at a cellular level in response to nutritional supplementation and exercise and its impact on muscle quality.

In healthy young men, HMB—a BCAA metabolite—can activate skeletal muscle protein synthesis and the main signaling pathways leading to protein synthesis [50,51]. In contrast, muscle in older patients lack the same adaptive response to increased loading compared to younger muscle, with decreased myofibrillar protein synthesis after resistance exercise [52,53]. A key reason may be the decline in mitochondrial function in skeletal muscle with age and the associated atrophy of myocytes [35]. Studies in mice with mitochondrial dysfunction demonstrate altered BCAA metabolism, promoting the catabolism of BCAAs to provide acetyl-CoA for de novo lipid synthesis upon metabolic stress [54]. This may explain the increase in intramuscular adipose tissue observed in our study, particularly in the older subgroup of patients. While this may seem counterintuitive, the increase in lipid content may not be an undesirable outcome in this patient population. Fatty acids derived from de novo lipid synthesis in skeletal muscle can subsequently be utilized in mitochondrial oxidation to provide energy for myocytes [55]. Furthermore, despite increased cellular lipid content in BCAA-treated myotubes, resistance to insulin—an inhibitor of muscle breakdown—was not altered [56]. Hence, we posit that, due to mitochondrial dysfunction in elderly skeletal muscle, there is a preferential storage of energy from HP ONS with HMB as lipids in the short term to meet the metabolic demands of resistance exercise and the physiological stress of surgery. This supports the increased role of HMB supplementation in the elderly, where its use in older adults may be more beneficial than in young adults.

For the increase in the IMAT index up to one-month post-surgery, it was likely that, in these older patients, adipose tissue was also the preferred source of energy to meet the body’s metabolic demands during and immediately after surgery. Surgery is a high stress event to the body—the resultant inflammation, corresponding with the extent of the surgical trauma, leads to a metabolic stress response and increased energy expenditure [57]. There is a catabolic response during surgical injury that results in increased triglyceride and free fatty acid breakdown [58]. To conserve glucose and meet elevated energy demands, adipose tissue becomes the major source of fuel postoperatively [59]. The HP ONS with HMB given to these patients could possibly have built a fuel reserve as intramuscular adipose tissue to help these patients with the energy expending process of surgical recovery, with a sustained increase in the IMAT index up to 1 month post-operatively. Subcutaneous fat, which is the typical energy storage when calories are in excess in the body and has been linked with poor outcomes [60,61], is distinct from IMAT in elderly patients. Our study provides interesting data suggesting that increasing the IMAT index in elderly sarcopenic patients during prehabilitation may instead be a sign of increasing reserves, in which intramuscular adipose tissue is an important reservoir of energy in the peri-operative period, rather than a marker of poor muscle quality.

### 4.3. Assessing Muscle Quality During Prehabilitation

To build muscle, training programs or prehabilitation regimes should typically last at least eight weeks to see effects on lean mass with protein supplementation [22,62,63]. However, the length of prehabilitation is usually limited by the risks of delaying surgery, particularly in curative oncologic cases. Due to the short prehabilitation period, appreciable change in muscle mass during this period is unlikely. The assessment of muscle quality may be more sensitive, detecting changes in muscle composition before changes in muscle size. This was observed in our study, where changes in the AI ultrasound-derived IMAT index occurred largely in the absence of changes in functional and anthropometric outcomes—besides the 6MWT and gait speed, which improved. While cross-sectional imaging with computed tomography (CT) and magnetic resonance imaging (MRI) are typically regarded as the gold standard for assessing muscle quality, they are prohibitive in terms of high cost, radiation exposure and the prudent utilization of resources [64,65]. Alternatively, the IMAT index can be assessed using ultrasound as a radiation-free point-of-care test with good inter-operator reliability [30]. Future studies may consider the use of the ultrasound-derived IMAT index, in conjunction with gait speed, to progressively evaluate patients’ response to prehabilitation and functional outcomes after surgery.

### 4.4. Limitations and Future Work

While our study offers insight on the altered muscle and adipose tissue metabolism in elderly sarcopenic patients and provides reassuring data on the maintenance of patients’ functional parameters before and after surgery, there are several limitations to be considered. The present study is a proof-of-concept one-arm pilot interventional trial with a small sample size. Future iterations would benefit from a larger study cohort and a control arm to elucidate the difference in muscle quality and functional outcomes without prehabilitation using HP ONS with HMB and resistance exercise. Secondly, we acknowledge limitations of the IMAT estimation equations by Young et al., which were developed by comparing ultrasound echo intensity with MRI percent fat [33]. The validations were carried out in a slightly younger adult population than our study; however, fat infiltration and muscle fibrosis are known to increase with aging and are difficult to differentiate by echo intensity [66]. Finally, our current results are limited by the short length of follow-up of only one month. Aside from functional parameters, additional outcomes such as patient satisfaction and quality of life scores provide valuable information when determining the optimal prehabilitation regime. In future work, we will investigate and discuss longer-term outcomes of up to six months post-operatively, including patient-reported outcome measures regarding their quality of life.

## 5. Conclusions

There was an increase in the IMAT index in sarcopenic patients after prehabilitation, which may be a result of altered muscle metabolism in elderly skeletal muscle. Functional outcomes were all at least maintained at baseline, with improvements seen in gait speed and the 6MWT. The utility of using ultrasound to track changes in the IMAT index during the peri-operative period should be further explored. There is promise in the implementation of a multimodal prehabilitation program incorporating HP ONS with HMB and resistance exercise. Longer-term outcomes, including quality of life measures, will be evaluated after the follow-up period is complete.

## Figures and Tables

**Figure 1 nutrients-16-04351-f001:**
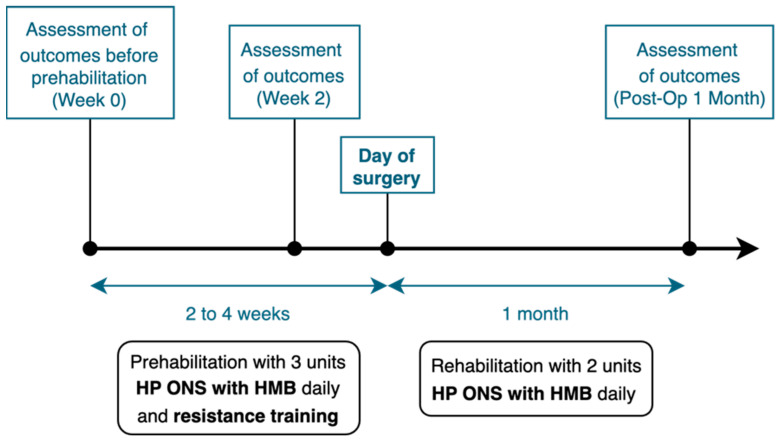
A patient’s peri-operative journey through the HEROS study.

**Figure 2 nutrients-16-04351-f002:**
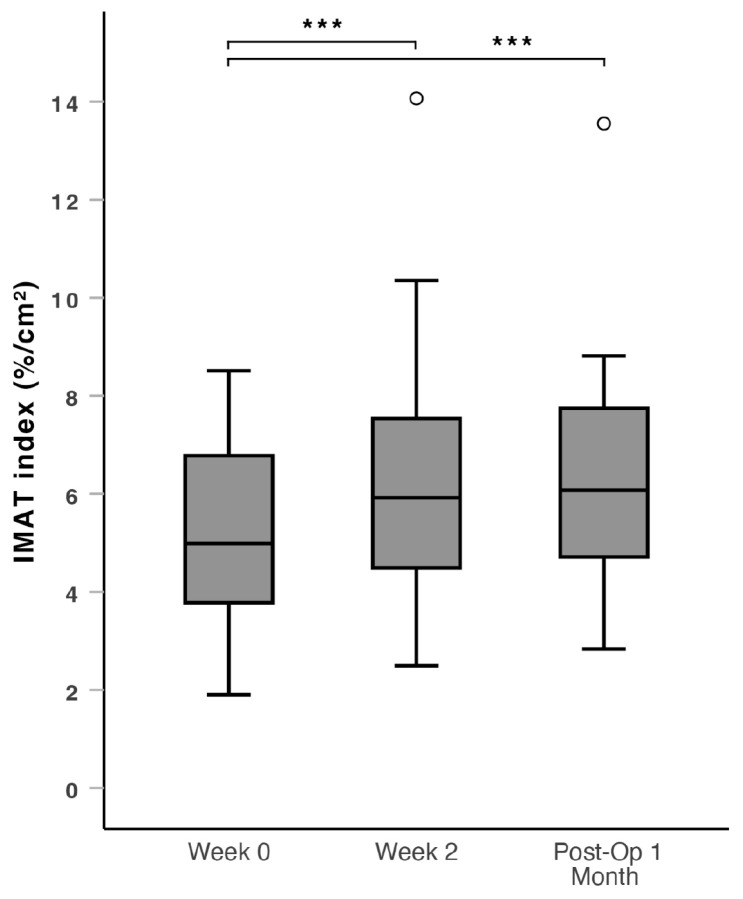
Box and whiskers plot of the IMAT index over the peri-operative window (Week 0, Week 2 and Post-Op 1 Month), which showed a significant increase from Week 0 to Week 2, that is sustained to Post-Op 1 Month. IMAT: intramuscular adipose tissue. ***: *p* < 0.05. ∘: outlier > 1.5 standard deviations from the mean.

**Figure 3 nutrients-16-04351-f003:**
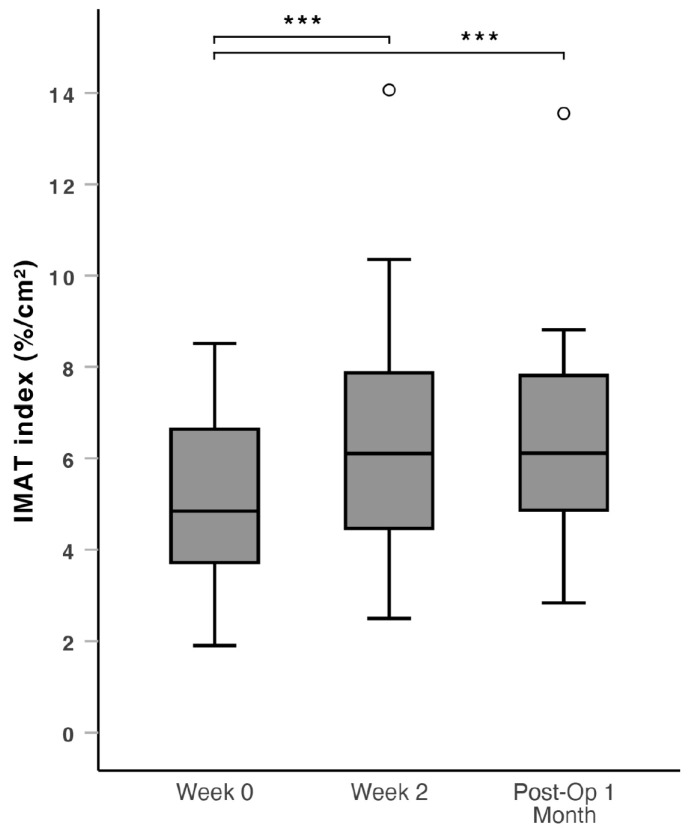
Box and whiskers plot of ultrasound-derived IMAT index over the peri-operative window (Week 0, Week 2 and Post-Op 1 Month) for the subgroup analysis of patients aged ≥ 65 years, which shows a significant increase after prehabilitation. IMAT: intramuscular adipose tissue. ***: *p* < 0.05. ∘: outlier > 1.5 standard deviations from the mean.

**Figure 4 nutrients-16-04351-f004:**
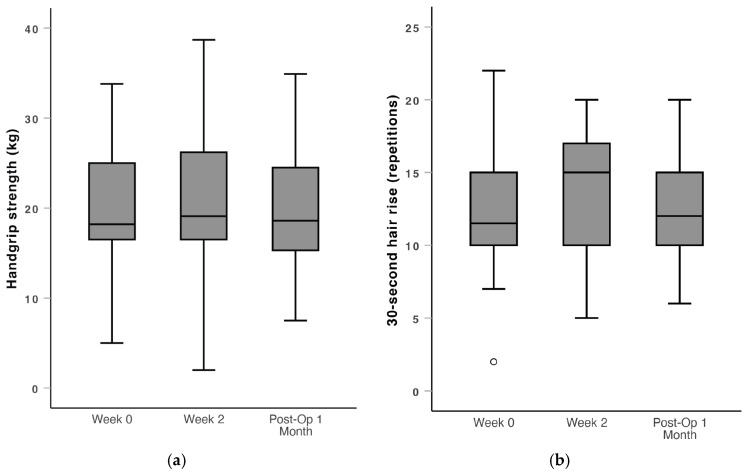
Box and whiskers plot of physiotherapist-measured outcomes over the peri-operative window (Week 0, Week 2 and Post-Op 1 Month). (**a**) Handgrip strength, (**b**) 30 s chair rise, (**c**) functional reach, (**d**) 6-min walk test, and (**e**) gait speed. 6-min walk test and gait speed both showed improvement from Week 0 to Week 2, before returning to baseline at Post-Op 1 Month. ***: *p* < 0.05. ∘: outlier > 1.5 standard deviations from the mean.

**Figure 5 nutrients-16-04351-f005:**
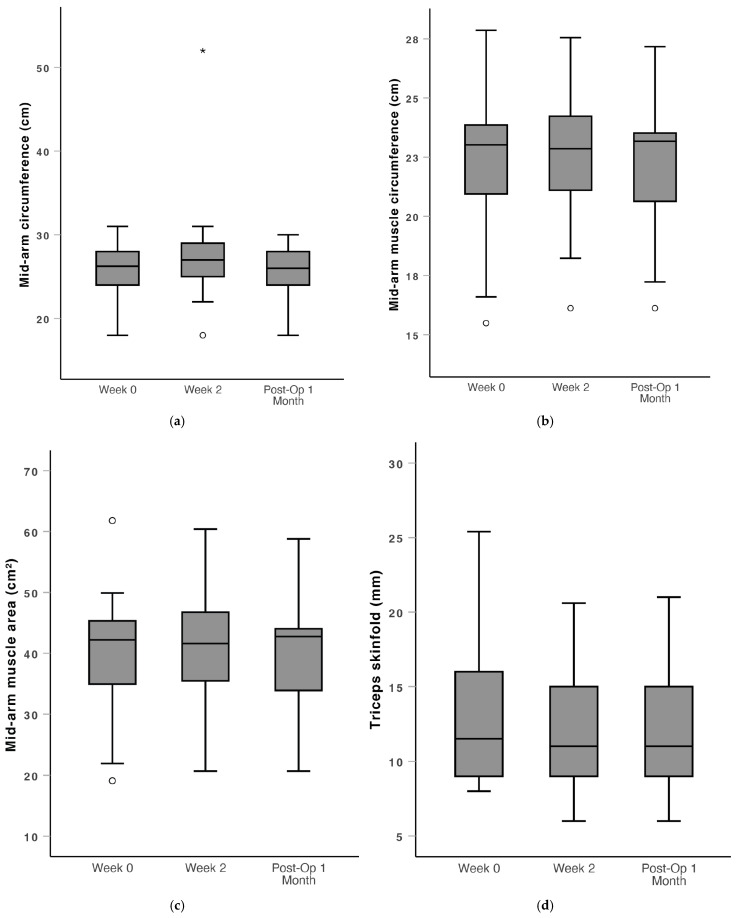
Box and whiskers plot of dietitian-measured outcomes over the peri-operative window (Week 0, Week 2 and Post-Op 1 Month), of which none were statistically significant. (**a**) Mid-arm circumference, (**b**) mid-arm muscle circumference, (**c**) mid-arm muscle area and (**d**) triceps skinfold. *: far outlier > 3 standard deviations from the mean. ∘: outlier > 1.5 standard deviations from the mean.

**Figure 6 nutrients-16-04351-f006:**
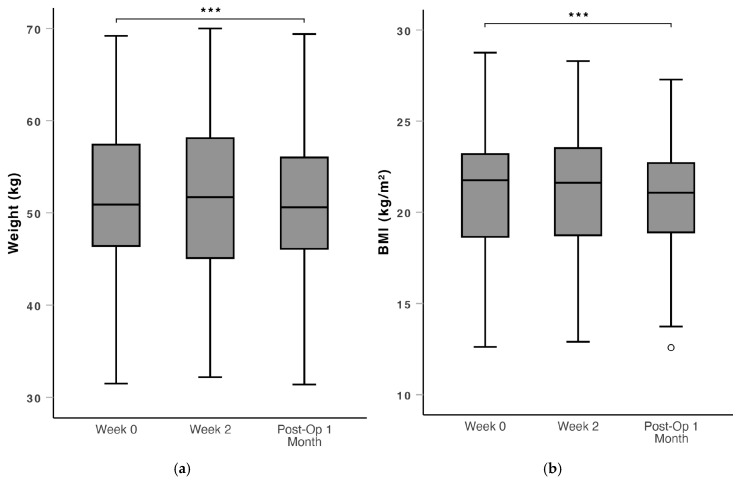
Box and whiskers plot of (**a**) weight and (**b**) BMI over the peri-operative window (Week 0, Week 2 and Post-Op 1 Month). While statistically significant, the actual change in weight and BMI are small and unlikely to be of clinical significance. BMI: body mass index. ***: *p* < 0.05. ∘: outlier > 1.5 standard deviations from the mean.

**Table 1 nutrients-16-04351-t001:** Macronutrient breakdown of Ensure^®^ Plus Advance. Three units daily (660 mL/day) were prescribed pre-operatively and two units daily (440 mL/day) were prescribed post-operatively. HMB: beta-hydroxy beta-methylbutyrate.

Nutrient	Per 100 mL	Per Unit (220 mL)
Energy (kcal)	150	330
Protein (g)	9.1	20.02
Fat (g)	4.8	10.56
Saturated fat (g)	0.45	0.99
Carbohydrate (g)	16.8	36.96
Sugars (g)	6.8	14.96
HMB (g)	0.55	1.21
Carnitine (mg)	18	40

**Table 2 nutrients-16-04351-t002:** Method of testing for each exercise test to assess functional parameters. Kg: kilograms; cm: centimeters; s: seconds; m/s: meters per second.

Exercise Test (Units)	Method of Testing
Handgrip strength (kg)	Measured using a handheld dynamometer The patient was instructed to squeeze as tightly as possible with their dominant hand, applying grip force
30 s chair rise (repetitions)	The patient was seated in the middle of the chair, feet shoulder-width apart and arms crossed, held against the chest The patient was instructed to complete as many full stands as possible within 30 s, sitting fully between each stand
Functional reach test (cm)	The patient was positioned standing next to a wall and with one arm positioned at 90 degrees of shoulder flexion with a closed fist The patient was instructed to reach forward as far as possible without taking a step The distance between the positions of the fists before and after reaching forward was measured
6 min walk test (m)	The patient was instructed to walk at their normal pace along an empty corridor Distance in meters that the patient covered in 6 min was recorded
Gait speed (m/s)	Derived from the 6-min walk test by dividing the distance covered in meters over 6 min (i.e., 360 s)

**Table 3 nutrients-16-04351-t003:** Study patient characteristics. BMI: body mass index.

Patient Characteristics	Total *n* = 36
Demographics	
Age in years, median (range)	71.5 (55–90)
Male sex, *n* (%)	18 (50%)
Race	
Chinese, *n* (%)	30 (83.3%)
Malay, *n* (%)	4 (11.1%)
Indian, *n* (%)	2 (5.6%)
Anthropometrics	
BMI (kg/m^2^), median (range)	21.6 (12.9–28.4)
Height (m), median (range)	1.56 (1.40–1.69)
Weight (kg), median (range)	51.0 (32.2–70.2)
Sarcopenia status	
Sarcopenia, *n* (%)	12 (33.3%)
Severe sarcopenia, *n* (%)	24 (66.7%)
Sarcopenia diagnostic measures	
Handgrip strength (kg), median (range)	18.1 (5.0–33.8)
6-m walk (m/s), median (range)	0.9 (0.17–2.58)
Bioelectrical impedance analysis (kg/m^2^), median (range)	5.76 (3.93–6.9)

**Table 4 nutrients-16-04351-t004:** Study outcomes at timepoints Week 0, Week 2 and Post-Op 1 Month reported as the median with standard deviation. SD: standard deviation; IMAT: intramuscular adipose tissue; cm: centimeter; kg: kilogram; reps: repetitions; m: meters; m/s: meters per second; mm: millimeters.

Outcome	Time		
	Week 0 (SD)	Week 2 (SD)	Post-Op 1 Month (SD)
**Muscle quality**			
IMAT%	12.8 (2.74)	13.0 (2.66)	13.3 (2.30)
IMAT index (%/cm^2^)	4.78 (1.63)	6.06 (2.50)	6.08 (2.11)
**Functional parameters**			
Handgrip strength (kg)	18.1 (6.11)	19.1 (6.81)	18.5 (6.58)
30 s chair rise (reps)	11.0 (4.78)	15.0 (4.78)	12.5 (4.52)
Functional reach test (cm)	20.5 (6.97)	23.5 (8.69)	20.0 (7.37)
6 min walk test (m)	345 (131)	387 (111)	328 (139)
Gait speed (m/s)	0.90 (0.44)	1.05 (0.31)	0.98 (0.29)
**Anthropometric measurements**			
Mid-arm circumference (cm)	26.0 (3.52)	27.0 (5.72)	26.0 (3.50)
Mid-arm muscle circumference (cm)	22.5 (2.87)	22.9 (2.39)	23.0 (3.00)
Mid-arm muscle area (cm^2^)	39.3 (9.57)	41.6 (8.39)	42.2 (10.0)
Triceps skinfold (mm)	10.0 (4.81)	11.0 (4.37)	11.0 (4.35)
Weight (kg)	50.8 (9.53)	51.0 (9.76)	51.2 (9.03)
BMI (kg/m^2^)	21.8 (3.57)	21.9 (3.52)	21.1 (3.32)

**Table 5 nutrients-16-04351-t005:** Healthy participant characteristics. BMI: body mass index.

Participant Characteristics	Total *n* = 10
Demographics	
Age in years, median (range)	62.5 (59–74)
Male sex, *n* (%)	4 (40%)
Race	
Chinese, *n* (%)	9 (90%)
Malay, *n* (%)	0 (0%)
Indian, *n* (%)	1 (10%)
Anthropometrics	
BMI (kg/m^2^), median (range)	24.4 (19.0–33.2)
Height (m), median (range)	1.63 (1.31–1.70)
Weight (kg), median (range)	58.0 (49.0–76.0)
Sarcopenia status	
Sarcopenia, *n* (%)	0 (0%)

## Data Availability

The data presented in this study are not publicly available due to institutional policy and can be made available upon request to the corresponding author.

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
