# Peer review of "Evaluating the Impact of Multimodal Prehabilitation with High Protein Oral Nutritional Supplementation (HP ONS) with Beta-Hydroxy Beta-Methylbutyrate (HMB) on Sarcopenic Surgical Patients—Interim Analysis of the HEROS Study"

_nutrients, 2024, doi:10.3390/nu16244351_

Round 1

Reviewer 1 Report

Comments and Suggestions for Authors

Major revisions:

Hong-Xiang et al examined the benefits of a multimodal intervention for muscle recovery following gastrointestinal surgery in sarcopenic patients. The study has significant weaknesses, which weakens its scientific merits, which is not only limited to not having a placebo group. 

As a start, please provide the following in the manuscript:

1. Complete breakdown of the resistance exercise program. This may be provided in the text in methods and should include the exercises, sets and repetitions, and progression. 

2. A table with a complete macronutrient breakdown of the supplements (Ensure Plus Advance, Vitamin, iron, and HMB) including daily dosing, kcals, protein, and fat.

3. Include the control group (n= 10) data in Table 2.

4. Include all diagnostic measures for sarcopenia in Table 2 as well to prove that your subjects were sarcopenic. 

5. Provide a full table showing the results of all outcomes with means +- standard error and/or deviations (day 0, two weeks and 1 month post).

You are missing a placebo control group and also have not provided a complete data breakdown of your baseline or post results. This makes it very difficult to publish. 

Reviewer 2 Report

Comments and Suggestions for Authors

The introduction is well-written, presenting the context and objectives clearly. However, the figures throughout the manuscript are not arranged in a well-organized layout. Improving their placement and alignment with the text could enhance the flow and clarity of the paper.

The Results section is presented very briefly, even though six figures are included. To improve readability and coherence, it might be beneficial to combine some of the results and streamline the presentation. Furthermore, the bar charts are all depicted in black and white, which makes it difficult to distinguish between different groups or categories. Using color or more distinct patterns could significantly enhance visual clarity and the reader's ability to interpret the data accurately.

In terms of language, while the writing is generally clear, it would benefit from greater variation in sentence structure to improve engagement and readability. For example, instead of repeatedly using phrases such as "xx is needed," alternative expressions could be employed to convey the same idea in more diverse ways, which would add depth and fluidity to the writing style.

In the section containing 259 words, I noticed what appears to be a typo. Carefully reviewing and revising this section for accuracy and precision would ensure a polished presentation.

Lastly, in the section with 281 words, the phrase "sth is used to derive speed" is ambiguous. It is unclear whether the author intends to convey that this process "is used to derive speed" or that it "will drive speed." Clarifying the intended meaning and revising the phrasing to reflect this clearly would enhance the reader's understanding and eliminate potential confusion. For example, if the intention is to emphasize an outcome, rephrasing it as "this method is expected to drive speed in the process" might be more precise.
